

# HgtSIM: a simulator for horizontal gene transfer (HGT) in microbial communities

Weizhi Song[1,2], Kerrin Steensen[1,3] and Torsten Thomas[1,4]

[1] Centre for Marine Bio-Innovation, University of New South Wales, Sydney, Australia
[2] School of Biotechnology and Biomolecular Sciences, University of New South Wales, Sydney, Australia
[3] Department of Genomic and Applied Microbiology, Georg-August Universität Göttingen, Göttingen, Germany
[4] School of Biological, Earth and Environmental Sciences, University of New South Wales, Sydney, Australia

## ABSTRACT

The development and application of metagenomic approaches have provided an opportunity to study and define horizontal gene transfer (HGT) on the level of microbial communities. However, no current metagenomic data simulation tools offers the option to introduce defined HGT within a microbial community. Here, we present HgtSIM, a pipeline to simulate HGT event among microbial community members with user-defined mutation levels. It was developed for testing and benchmarking pipelines for recovering HGTs from complex microbial datasets. HgtSIM is implemented in Python3 and is freely available at: https://github.com/songweizhi/HgtSIM.

# INTRODUCTION

Horizontal gene transfer (HGT) has been recognized as an important force in microbial evolution and adaptation (*Soucy, Huang & Gogarten, 2015*). A number of pipelines have been developed to identify HGTs in draft or completed genomes of isolated microorganisms (*Adato et al., 2015*; *Hasan et al., 2012*; *Podell & Gaasterland, 2007*; *Ravenhall et al., 2015*; *Trappe, Marschall & Renard, 2016*; *Zhu, Kosoy & Dittmar, 2014*). In recent years, the development and application of metagenomic approaches have provided novel and vast amounts of information on the genomic composition of uncultured microorganisms (*Thomas, Gilbert & Meyer, 2012*). This offers an opportunity to study HGT on the level of microbial communities, however new bioinformatics tools and pipelines have to be developed to reliably detect any HGT events in metagenomic datasets. Simulations of metagenomics reads have been essential for the development and benchmarking of pipelines for the quality control, assembly and annotation of metagenomic data (*Peng et al., 2012*; *Kang et al., 2015*). These simulation tools typically produce reads based on defined sets of reference genomes with user-defined abundance distributions and often considering realistic error models for common sequencing technologies (*Escalona, Rocha & Posada, 2016*). However, no current simulation tool offers the option to introduce defined HGT within the microbial community data simulated, thus allowing to test pipelines that

Corresponding author
Torsten Thomas,
t.thomas@unsw.edu.au

aim to detect HGT. Here, we have developed a pipeline called HgtSIM, which can simulate HGTs between the genomes of microbial communities. The pipeline can simulate HGTs with different degrees of similarity for transferred genes found in donor and recipient genomes, thus allowing to assess the detection of relatively recent or past transfers.

## METHODS

### Simulation of gene mutations

The transfer of genes into a recipient genome often involves subsequent mutations that reflect evolutionary drift or adaptation to the new genomic context (e.g., change in codon usage to match tRNA availability). To simulate such mutations without disrupting reading frames and to confine the mutations to a defined range, we use codons as units of mutations. The mutations of codons were grouped into four categories ($C_i$): (1) one-base, silent mutation; (2) one-base, non-silent mutation; (3) two-bases mutations and (4) three-bases mutations (Table 1).

The algorithm for simulating random mutations is as follows:

(1) Get the length ($L$) of each gene to be transferred.
(2) Define the number of bases need to be changed ($N$) based on a user-defined identity value ($I$) and $L$: i.e., $N = LI/100$.
(3) Define the type of mutations based on $N$ and a user-defined ratio of the four mutation categories. For example, if a ratio of 1:1:1:1 is specified for $C_1$:$C_2$:$C_3$:$C_4$, then, $N = C_1 + C_2 + 2C_3 + 3C_4$.
(4) Randomly select $C_1, C_2, C_3$ and $C_4$ codons and perform the corresponding mutations.

All changed nucleotides are recorded in a mutation report file. A BlastP-based comparison between the amino acid sequences is also provided.

### Simulation of gene transfers

The steps to simulate random gene transfers are as follows (Fig. 1):

(1) Add flanking sequences (if specified) to the (mutated) genes to be transferred. These flanking regions could, for example, be transposon insertion sequences.
(2) Get the total length or the total number of intergenic regions of the recipient genome ($P$) and user-defined number of genes ($Q$) to be transferred.
(3) Randomly select $Q$ numbers between 1 and $P$ and cut the recipient genome at corresponding positions to create sub-sequences. If user wants to insert gene transfers only into intergenic regions, then the recipient genome will be cut in the middle position of the selected intergenic regions.
(4) Randomly assign the (mutated) genes to be transferred to the cut points and concatenate them with the sub-sequences.

All the break positions and the (mutated) genes inserted to these positions are recorded in an insertion report file.

The Python3 implementation of this HgtSIM algorithm, parameter setting and all scripts used here are available at: https://github.com/songweizhi/HgtSIM.
**Table 1  Mutation types of codons.** The changed bases are displayed in bold. The corresponding amino acid change is given in parenthesis. As the number of silent two- and three-bases mutations are low (1%) compared to non-silent mutations, we here combined them into the same categories. The start and stop codons were excluded when calculating the number of mutation types.

| Category | Mutation type | Example | Total number |
|---|---|---|---|
| $C_1$ | One-base, silent | AT**C** (Ile) → AT**A** (Ile) | 124 |
| $C_2$ | One-base, non-silent | **G**CC (Ala) →**A**CC (Thr) | 356 |
| $C_3$ | Two bases, silent | **A**G**G** (Arg) →**C**G**T**(Arg) | 20 |
|  | Two bases, non-silent | C**TC** (Leu) → C**CT** (Pro) | 1,394 |
| $C_4$ | Three bases, silent | **AGT** (Ser) →**TCC** (Ser) | 12 |
|  | Three bases, non-silent | **GTG** (Val) →**TAC** (Tyr) | 1,400 |

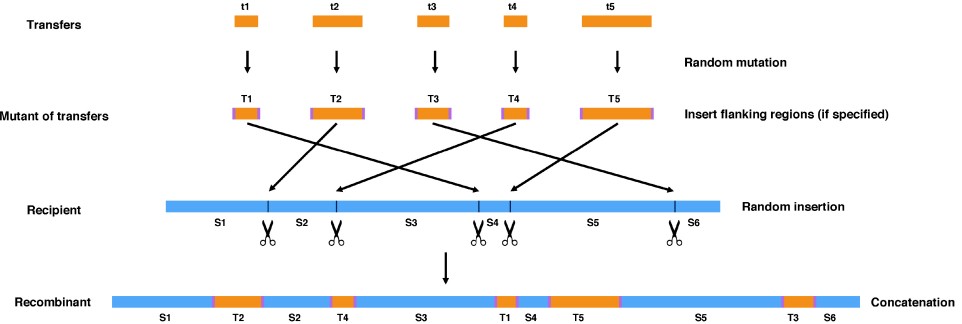

**Figure 1    The workflow of HgtSIM.**

## RESULTS AND DISCUSSION

### The effect of mutation categories on the level of coded amino acid changes

The correlation of mutation on the nucleotide level and the resulting amino acid changes under different ratios of mutation categories were assessed by performing random mutations on 100 genes selected from ten Alphaproteobacteria genomes (Table 2). The values for the level of user-defined nucleotide mutations and the values for the resulting changes in protein sequences were similar for the category ratios of ''0:0:0:1'' and ''1:0:1:1'' (Fig. 2). This correlation analysis provides the user with information on the level of protein sequence changes that occur at any given nucleotide mutation level and category settings.

### The effect of assemble k-mer range on the recovery of simulated HGTs

We next demonstrated the usefulness of HgtSIM to assess the recovery rate of HGTs from a simulated metagenomic shotgun-sequencing dataset after various sequence assembly processes. For this, 10 genes each were selected from the ten Alphaproteobacteria genomes and randomly transferred to ten Betaproteobacteria genomes (Table 2) with various degrees of mutation (0%, 5%, 10%, 15%, 20%, 25% and 30%). The ratio of mutation types was set to 1:0:1:1 and a flanking sequence of ''TAGATGAGTGATTAGTTAGTTA''

**Table 2  The selected 20 genomes used in this study.**

| Class | Strain | NCBI BioProject ID | Genome size (Mbp) |
|---|---|---|---|
| | *Acidiphilium multivorum* AIU301 | 60101 | 3.58 |
| | *Ketogulonigenium vulgarum* WSH 001 | 161161 | 2.64 |
| | *Mesorhizobium australicum* WSM2073 | 47287 | 3.74 |
| | *Methylocapsa acidiphila* B2 | 72841 | 5.91 |
| | *Methyloferula stellata* AR4 | 165575 | 4.04 |
| *Alphaproteobacteria* | *Rhodovibrio salinarum* DSM 9154 | 84315 | 4.30 |
| | *Roseobacter litoralis* Och 149 | 19357 | 3.98 |
| | *Sphingobium japonicum* UT26S 1 | 19949 | 3.35 |
| | *Starkeya novella* DSM 506 | 37659 | 4.54 |
| | *Tistrella mobilis* KA081020 065 | 76349 | 3.74 |
| | *Alicycliphilus denitrificans* K601 | 50751 | 4.76 |
| | *Dechlorosoma suillum* PS | 37693 | 3.63 |
| | *Gallionella capsiferriformans* ES 2 | 32827 | 3.02 |
| | *Herbaspirillum seropedicae* SmR1 | 47945 | 5.26 |
| *Betaproteobacteria* | *Nitrosospira multiformis* ATCC 25196 | 13912 | 3.04 |
| | *Ramlibacter tataouinensis* TTB310 | 16294 | 3.88 |
| | *Sideroxydans lithotrophicus* ES 1 | 33161 | 2.41 |
| | *Snodgrassella alvi* wkB2 | 167602 | 2.99 |
| | *Sulfuricella denitrificans* skB26 | 170011 | 2.86 |
| | *Tetrathiobacter kashmirensis* WT001 | 67337 | 4.16 |

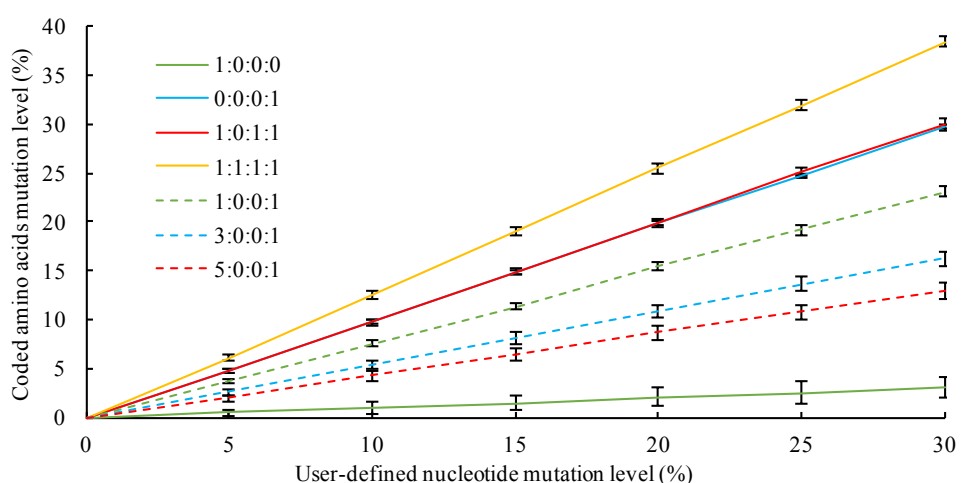

**Figure 2  The correlation of mutation on the nucleotide level and the resulting aa changes under different mutation category ratios.** The four numbers separated by colon refer to the ratio between $C_1, C_2, C_3$ and $C_4$.

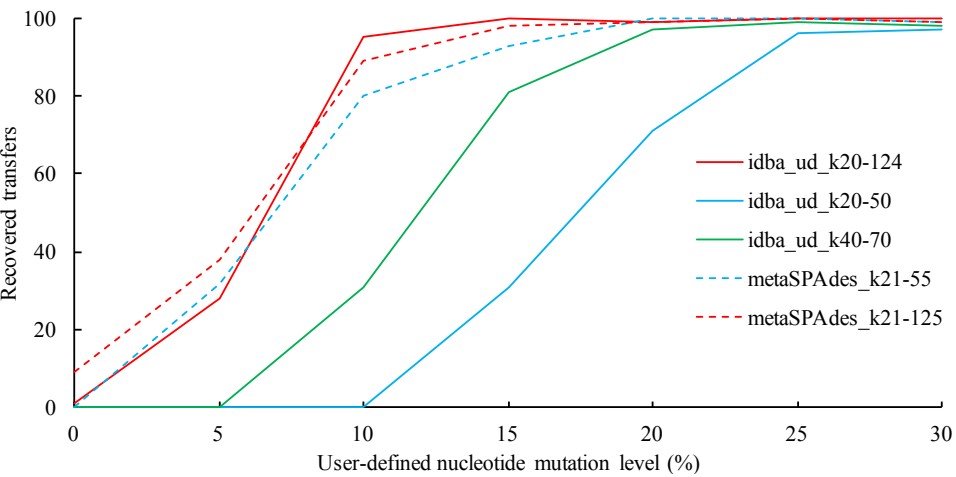

**Figure 3** **The effect of assemble k-mer range on the recovery of HGT events.**

were added to the two ends of all transfers. Ten million paired-end, error-free 100-bp reads (corresponding to a coverage of 26.4×) with 250 bp insert size were simulated with an in-house script from the 20 genomes for each mutation group. To get an even sequencing depth distribution of the 20 genomes, their relative abundances were all set to one. The simulated reads were then assembled with IDBA_UD 1.1.1 (*Peng et al., 2012*) and metaSPAdes 3.9.0 (*Nurk et al., 2017*) with multiple k-mer ranges (Fig. 3). A gene transfer was considered to be recovered during the assembly if at least one of the gene's two flanking regions was >1 Kbp and the flanking region matched its recipient genome. To do this, a blastn (*Altschul et al., 1990*) was performed between the introduced gene transfers and the contigs produced by the assemblers. The blast results were then filtered with an identity cutoff of 99% and a coverage cutoff of 99% for the transferred genes.

The results show that the best recovery was obtained with a k-mer range of 20–124 for IDBA_UD as well as 21–55 and 21–125 for metaSPAdes (Fig. 3). The number of genes recovered by the two assemblers were reduced when the user-defined nucleotide mutation levels were low (i.e., <5%). When no mutation was introduced, only one and nine genes were recovered by IDBA_UD and metaSPAdes, respectively.

**The effect of sequencing depth on the recovery of no-mutation HGTs**
We then investigated how sequencing depth might affect the recovery of HGTs with no mutations. To do this, between one to 20 million paired-end 100-bp reads with 250 bp insert size were simulated for the 20 genomes, no error was introduced to the simulated reads during simulation and no mutation was introduced to the 100 transferred genes. The simulated reads were then assembled with IDBA_UD and metaSPAdes with the optimal k-mer ranges identified above. Assembly statistics (total length, number of contigs, N50 and percentage of recovered reference genomes) were obtained with MetaQUAST 4.5 (*Mikheenko, Saveliev & Gurevich, 2015*).
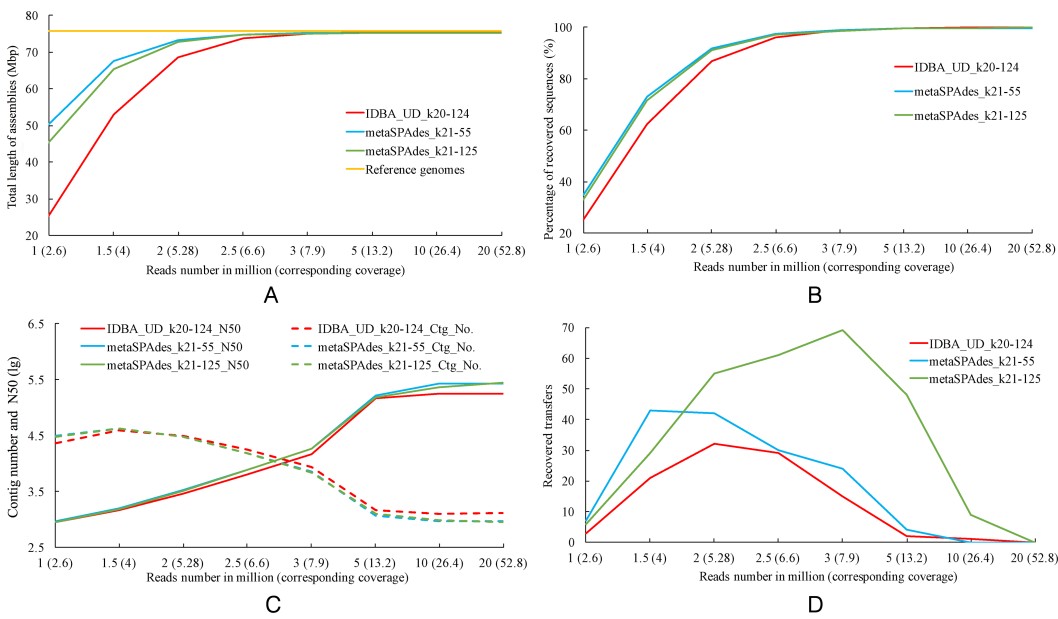

**Figure 4** **The total length (A), percentage of recovered sequences (B), contig number and N50 (C) of assembler produced assemblies. (D) Number of recovered transfers.** The lines showing the number of contigs and N50 of metaSPAdes produced assemblies with two different k-mer settings were overlapping in panel (C).

The results show that the quality of assemblies improved with increasing sequencing depth (Figs. 4A–4C) and over 98.5% of sequences for the reference genomes were reconstructed with sequencing depths of greater 6.6× (Fig. 4A). We found that the number of gene transfers recovered by IDBA_UD and metaSPAdes was not linearly correlated with sequencing depth. The best recovery (69 out of 100 transfers) was observed with metaSPAdes at a k-mer range of 21–125 and sequencing depth of 7.9× (Fig. 4D). With a k-mer range of 21–55, 43 gene transfers were recovered by metaSPAdes when the sequencing depth is 4×. As for IDBA_UD, the best recovery was obtained with a sequencing depth of 5.28×. The decrease of HGT recovery rates beyond a certain coverage threshold was surprising and contrasted the improved general quality measurements of the assemblies (Figs. 4A–4C). One possible explanation is that assemblers under certain coverage condition are more likely to assemble contigs for the transferred genes that lack flanking regions, and visual inspection of assembly graphs found instances of this.

## The effect of read length and insert size on the recovery of no-mutation HGTs

We also simulated how insert size and read length might influence recovery of transfer events. As the best recovery of no-mutation HGTs was observed with metaSPAdes and a k-mer range of 21–125 at sequencing depth of 7.9× (Fig. 4D), we simulated reads with different length (100 bp and 250 bp) and insert sizes (250 bp, 500 bp and 1 Kbp) to this depth. More no-mutation gene transfers were recovered with reads length of 100 bp than

**Table 3    The effect of reads length and insert size on the recovery of 100 simulated HGT events.**

| Reads length (bp) | 100 | | | 250 | | |
|---|---|---|---|---|---|---|
| Insert size (bp) | 250 | 500 | 1,000 | 250 | 500 | 1,000 |
| Recovered gene transfers | 69 | 55 | 63 | 15 | 23 | 51 |

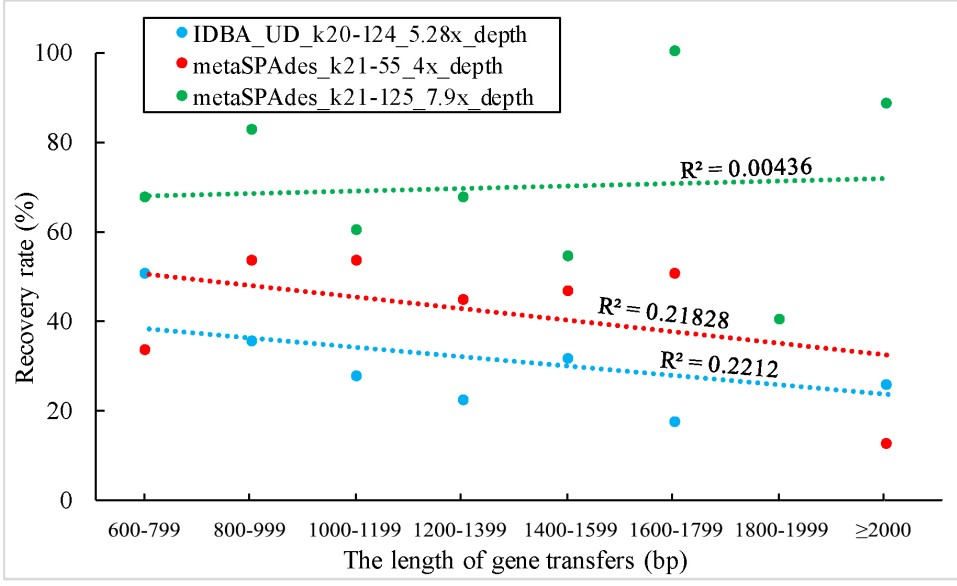

**Figure 5    The correlation between the length of gene transfers and their recovery rate.**

with 250 bp. For the datasets with 250 bp read length the recovery of gene transfers was improved with longer insert sizes (Table 3).

## The effect of the DNA length on the rate of transfer recovery

The correlation between the length of the DNA transferred and its recovery rate was also analyzed. The three datasets shown in Fig. 4D were used for this analysis. There was no statistically supported correlation between the gene length and the recovery rate, indicating that gene length has no impact on recovery rate under the given experimental conditions (Fig. 5).

## CONCLUSIONS

Our study demonstrates how various aspects of metagenomic sequencing projects (e.g., insert length, read length, assembly parameters, gene length) can influence the potential to recover HGT from metagenomic datasets. Testing and benchmarking of various parameters and tools with simulated datasets produced by HgtSIM will in the future help to develop robust pipelines that have maximal success in recovering HGT from complex metagenomic data.

### Funding

This research was funded by the Australian Research Council. Weizhi Song was funded by the China Scholarship Council (201508200019). The funders had no role in study design, data collection and analysis, decision to publish, or preparation of the manuscript.

### Grant Disclosures

The following grant information was disclosed by the authors:
Australian Research Council.
China Scholarship Council: 201508200019.

### Competing Interests

Torsten Thomas is an Academic Editor for PeerJ.

### Author Contributions

- Weizhi Song conceived and designed the experiments, performed the experiments, analyzed the data, contributed reagents/materials/analysis tools, wrote the paper, prepared figures and/or tables, reviewed drafts of the paper.
- Kerrin Steensen conceived and designed the experiments, performed the experiments, analyzed the data, contributed reagents/materials/analysis tools, prepared figures and/or tables, reviewed drafts of the paper.
- Torsten Thomas conceived and designed the experiments, analyzed the data, wrote the paper, reviewed drafts of the paper.

### Data Availability

GitHub: https://github.com/songweizhi/HgtSIM.

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
