# Peer review of "HgtSIM: a simulator for horizontal gene transfer (HGT) in microbial communities"

_PeerJ, doi:10.7717/peerj.4015_

## Round 0.1 · original submission · Major Revisions

Importantly, please add sufficient background information to the introduction, including references for most recent tools for HGT detecting. Along these lines, it would be warranted to apply one of the HGT detection tools to the simulated data (see reviewer #3) and also carefully discuss how the simulated data might diverge from real biological data and how far one might be able to extrapolate from these results to real data (see also reviewer #2).

As for the tool itself, several deficiencies were identified, which need to be addressed, including its utility beyond 1-contig assemblies, the inclusions of an optional "mixed mode", making adjustments for flanking sequences (varying sequence length and composition) to better simulate true biological data, etc (for more details see reviewer #1). Please

Further, please adhere to the journal specifications for publications on “bioinformatics software tools” https://peerj.com/about/policies-and-procedures/#discipline-standards (see bullet #2), as this is the category your manuscript would fall under.

Reviewer 1 ·

Basic reporting

no comment

Experimental design

no comment

Validity of the findings

no comment

Comments for the author

Song et al. describe the open-source tool HgtSIM, which was developed to simulate HGT events among microbial community members taking into account user-defined mutation levels. The authors give an illustrative example how HgtSIM works, starting from the insertion of a set of selected genes into a collection of genomes, followed by modification of these genes simulating mutation and adaption. Song et al. then employ the tool GemSIM to simulate metagenomic reads from their simulated dataset, perform assemblies using the assembler IDBA-UD and finally assess the recovery rate of gene transfer events.

HgtSI is not the first tool to simulate HGT in microbial genomes (see HGT options of EvolSimulator by Beiko&Charlebois, 2007, https://doi.org/10.1093/bioinformatics/btm024), but seems to be more straight-forward to use. It could be downloaded from Github together with a comprehensive documentation. HgtSIM is written in python 3 and the code is well readable.

After testing HgtSIM I felt that it could be slightly more flexible in terms of query genome input format and user-specified options seemed to be rather limited.

HgtSIM was successfully used on the provided test dataset. However, gene insertion was only possible on closed genomes (or single contigs), as soon as additional contigs were added (in the same file) the HGT simulation failed. This was somewhat surprising as the tool supposedly was designed to simulate HGTs in metagenomes - many published metagenome assembled genomes are not closed. Thus, it would be great if HgtSIM could also be used on genomes with more than 1 contig.

Flanking sequences are often introduced together with transferred genes. In HgtSIM the user has the option to provide upstream and downstream flanking sequences, which is great. However, it seems that these flanking sequences are then equally used for all inserted genes. I would suggest to implement a feature to automatically generate flanking sequences with varying sequence length and composition (within a given range). This would help to make the simulation result more similar to real world data.

An important feature of HgtSIM is that acquired genes do undergo evolutionary drift and adaptation to the new cellular context. However, in its current form this process felt slightly limited due to the fixed identity value provided by the user. Ideally, HgtSIM would include an optional "mixed mode" - which would use a range of different identity values randomly to alter the query genes.

Line 119: The here described results imply that recent HGT is most difficult to detect. I was wondering to what extend this observation was affected by the assembler (IDBA-UD) and to what extend a better performing assembler would change the result. An alternative assembler could be SPAdes (or metaSPAdes), it is popular and widely-used and presumably outperforms IDBA-UD considering contig and scaffold lengths (see Vollmers et al., 2017 https://doi.org/10.1371/journal.pone.0169662).

Lastly, the authors evaluated the impacts of read length and insert size on the recovery rate but they did not elaborate on the potential influence of gene length. I would assume that successful detection negatively correlates with gene length. If there is such correlation, then a length distribution plot of inserted genes together with their recovery rate would be a helpful reference for the user.

Reviewer 2 ·

Basic reporting

The manuscript presents a software for the simulation of genomes, which contain simulated horizontally transferred genes. The authors provide an example application to an artificial microbial community. I agree this is an interesting aspect in metagenomics, which besides strain diversity and limited coverage introduces an additional challenge to metagenome assemblies.

The manuscript is a brief “Applications Note” rather a full research article, as required for PeerJ. It lacks a central research question and suitable, comprehensive experiments to address it. For example, the manuscript does not introduce the challenges of metagenomic assemblies in detail, and does not address the problem of HGT event prediction, or donor/acceptor inference, for which HGT simulations might also be useful. The results on an artificial set of genomes, using one assembler, demonstrate the general utility of HgtSIM and give some general hints for metagenomics assembly. For those who work on shutgun metagenomes, the results are quite expected. However, they don’t resemble any realistic microbial community and thus have no meaning for metagenomic assembly practice.

Experimental design

I have downloaded and tested the software. It is written in Python3 and has only few dependencies, which are easy to fulfil. I did not completely understand the implementation strategy, as parts of the script seem unnecessary (e.g., the genetic code is hardcoded although the script imports Biopython, which already provides more sophisticated genetic code classes and methods). Anyway, I could successfully run a simple test, based on the provided example files.

Validity of the findings

Does not apply.

Comments for the author

To become a PeerJ Research Article, the manuscript would require fundamental expansion. A suitable alternative might be a submission to OUP Bioinformatics, as “Applications Note”.

Reviewer 3 ·

Basic reporting

The authors present a simulator for horizontal gene transfer in microbial communities.
The article is written in unambiguous and professional English and only a few typos need to be corrected.
I would suggest to add more background into the introduction and also include a couple of references about more recent tools for detecting HGT, e.g.
https://doi.org/10.1371/journal.pcbi.1004408
https://doi.org/10.1371/journal.pcbi.1004095
https://doi.org/10.1093/bioinformatics/btw423

Figure 1B is hard to read. It is impossible to distinguish some of the labels. Please refine.

In Table 2 it would be useful to add the size of the genomes, as this has an effect on the simulated coverage (see experimental design), which again might have an effect on the results presented in chapters 3.2 and 3.3.

Table 3 is missing a caption which explains the numbers in the table. I guess these are recovery rates (in %) of the HGT genes. Also, the 0% and 5% need to be explained in the caption or table.

The authors do not share the raw data, but at least provide accessions for the genomes used for the simulation. Please provide all details about which parameters were used with the different tools (read simulator and IDBA_UD), so that results are reproducible.

Experimental design

In general, it remains unclear what the goal of the simulation should be. What should be benchmarked with the simulated data? Is it mainly about assembly performance? If so, the authors have to make sure that they are not measuring a different effect when evaluating the assembly (see below). Or is the goal to benchmark tools for detection of HGT? If so, it would be nice to see how one if these tools performs. I understand that a thorough benchmarking of HGT detection tools might not be within the scope of this manuscript, but if there are benchmarks published already, the “best tool” could be applied on the simulated data set.

Lines 54-62: Taking a number of gene sequences as input and inserting them into a number of target genomes seems to be a relative straightforward task. Having an automated way to do so is very useful for simulation and benchmarking. Therefore, the described pipeline is a step into the right direction, if such studies are performed often and in high-throughput. It remains unclear, though, how realistic the chosen model of mutating the horizontally transferred genes is. The authors barely explain the underlying principles of their model. One can immediately argue, that a distinction between one-base (silent or non-silent), two-base and three base mutations is too simplistic. There are examples where two-base or even three-base mutations are silent mutations (e.g. UCU and AGC both code for Serine). There are more sophisticated models available, could they be integrated in the simulation pipeline (e.g. https://doi.org/10.1080/10635150701546231)?

Line 71: Even for the simple model, how would one choose C1:C2:C3:C4? What is the default for the pipeline and why was it chosen this way?

Line 80: Does the recipient genome have to be a finished genome (i.e. no gaps allowed)? Many genomes are permanent draft genomes which cannot be used for a large scale simulation if the pipeline only works for finished genomes.

Line 84: When cutting the genome for inserting the HT genes, this will disrupt genes in the recipient genome? Is this realistic assumption or are insertions into CDSes much less likely? If so, this should be included in the model

Line 100: “The category ratios of “0:0:0:1” and “1:0:1:1” resulted in level of amino acid sequence changes that were similar to the user-defined level of nucleotide mutations” This sentence is unclear. What exactly is the user-defined level of nucleotide mutations? And how does this relate to these ratios?

In general, details about running the pipeline and assembly (parameter setting) are missing, which makes reproducibility difficult.

Validity of the findings

The authors present a small benchmark data set consisting of 20 genomes, where 100 genes from Alphaproteobacteria get randomly transferred to 10 genomes of Betaproteobacteria. I am not convinced that the effects of missing genes in the subsequent assemblies are actually always the degree of mutations. If we assume that the 20 genomes have a length of 4Mbp on average, the sequencing depth for the simulated data set is only about 12.5x (10 mio x 100 bp / 20 * 4 Mbp = 12.5). This is already a pretty low coverage for assemblies based on long kmers and could explain that for kmer ranges >60 most of the genes are lost. What are the total assembly sizes for these assemblies? I would guess they are rather small. Additionally, a couple of genomes are >5Mbp long, which reduces the coverage for these genomes even further.
Line 120: therefore, the mink parameter has a general effect on the assembly of low coverage genomes and not on HGT detection. Only using show kmers (20) in the kmer range pick up the low coverage genomes and therefore allow HGT detection

Line 131: there are no results shown for longer reads of different input sizes

I would suggest to prepare two different data sets, one without HGT and one including HGT. Comparing these two datasets would elucidate if the missing genes are actually an effect of HGT or just an effect of too low sequence coverage.

Comments for the author

The manuscript seems to be very preliminary and needs a more carefully designed experimental setup. It is not clear, if the findings are an effect of HGT or missing sequencing depth. Also, it would be nice to see an HGT detection tool applied on the data set.

---

## Round 0.2 · Major Revisions

As is, this manuscript is still not acceptable for publication with Peer J and I urge the authors to thoroughly address the concerns raised by reviewer #3.

As pointed out by reviewer #3, the methods are still not described in proper detail and the presented results not reproducible. Please provide all data, scripts, methods details, etc. needed to enable reproducibility of this study.

Please also elaborate on and better discuss and interpret your result (for more details see #3).

We hope you are willing to address these concerns and resubmit your work.

Reviewer 1 ·

Basic reporting

no comment

Experimental design

no comment

Validity of the findings

no comment

Comments for the author

no comment

Reviewer 2 ·

Basic reporting

no comment

Experimental design

no comment

Validity of the findings

no comment

Comments for the author

The revised version has been substantially improved. The authors have carefully addressed the reviewer's comments. As there is not any substantial issue remaining from my side, I thank the authors and recommend acceptance of the manuscript.

Reviewer 3 ·

Basic reporting

The authors switched to metagenome assemblers for the simulation study.
This improved the quality of the simulation study tremendously. However, the findings
are not reproducible, if details of the simulation are not described. This is the case for
many details in the study and need to be addressed (see details below).

Please correct the following typos:
line 29: ...tool offers...
line 31: ...events...
line 49: ...error...
line 124: ...paired-end...
line 144: ...optimal...
line 163: ...read length...

Figure 4A, y-axis: Assemblies

Experimental design

lines 122-124: which mutation model were used for the simulation?
Were flanking regions added to the transfered genes? If so, what sequences, how long?

lines 124-126: does the script introduce sequencing errors?
Please add the resulting sequencing coverage to the main text.

The methods are still not described in detail. The presented results are not reproducible.
Please provide the all input files for the scripts such that the study can be reproduced.
The GitHub repository has all the necessary scripts, but e.g. the abundance.txt file or
the input_genomes_m0 files are missing.

How exactly were recovered genes identified? Alignment against the reference?
Which tool was used? What were the alignment parameters? And what the criteria for
calling a HGT gene recovered?

I found all details in the python scripts on GitHub, but they should be included in the manuscript (Methods section). Also, please include the version numbers of the assemblers.

Validity of the findings

line 135: what happens to the HGT genes? Are they not assembled at all? Do you recover only one copy of the gene?

lines 146-152: First, it is not surprising that the recovery rate is not increasing linearly with sequencing depth. It is well known, that the assembly in terms of recovered genome sequence behaves in such a way (Lander and Waterman 1988).

Very surprising is the fact, that higher sequencing coverage leads to a lower recovery rates. I was convinced that there is a bug in the simulation and analysis pipeline, but the results seem to be correct.

I evaluated the assemblies using metaQUAST and from these results,
the higher coverage data sets seem to produce better assemblies (less contigs, higher genome recovery, higher N50, etc). I would suggest to add these basic statistics to the manuscript, just to show that there are no flaws in the simulation process.

Taking a closer look why the HGT genes get lost in the high coverage data sets, I was able to re-run the analysis using the Assembler_recovered_transfers_iden100.py script from GitHub and the data provided. It was very surprising for me to see that the flanking regions get lost in the assemblies with higher coverage.

Here is an excerpt from the BLASTN results generated by the script for
the metaspade_m0_k21-125_l100i250_3_million.fasta assembly

AMAC_01196 NODE_75_length_51023_cov_1.46669 100.00 1203
AMAC_02518 NODE_2577_length_11107_cov_1.63078 100.00 1269
AMAC_02914 NODE_5740_length_2456_cov_2.52716 100.00 1908
AMAC_03303 NODE_3040_length_9082_cov_1.74892 100.00 1452
AMAU_01212 NODE_938_length_22333_cov_1.55155 100.00 1530
AMAU_01255 NODE_1152_length_20072_cov_1.53219 100.00 1497
AMAU_02414 NODE_3750_length_6844_cov_1.81192 100.00 1290
AMAU_02488 NODE_4885_length_4137_cov_2.12155 100.00 1518
AMAU_04187 NODE_921_length_22539_cov_1.54172 100.00 1176
AMS_00102 NODE_26_length_64149_cov_1.46827 100.00 1395
AMS_01465 NODE_491_length_29719_cov_1.50854 100.00 1413
AMS_01716 NODE_1033_length_21171_cov_1.56804 100.00 1176
AMS_01785 NODE_3328_length_8142_cov_1.76752 100.00 1194
AMS_02653 NODE_869_length_23152_cov_1.47987 100.00 1062
AMS_02811 NODE_139_length_44053_cov_1.48303 100.00 1410

For the higher coverage data set (assembly metaspade_m0_k21-125_l100i250_20_million.fasta) the BLASTN results look like this:

AMAC_00685 NODE_684_length_1608_cov_19.649 100.00 1608
AMAC_01196 NODE_766_length_1203_cov_20.8583 100.00 1203
AMAC_02518 NODE_752_length_1269_cov_17.6364 100.00 1269
AMAC_02914 NODE_660_length_1908_cov_18.9573 100.00 1908
AMAC_03303 NODE_711_length_1452_cov_17.7535 100.00 1452
AMAU_01212 NODE_697_length_1530_cov_19.0856 100.00 1530
AMAU_01255 NODE_702_length_1497_cov_18.6773 100.00 1497
AMAU_02414 NODE_744_length_1291_cov_18.9364 100.00 1290
AMAU_02488 NODE_700_length_1521_cov_19.2229 100.00 1518
AMAU_04187 NODE_792_length_1102_cov_28.4966 100.00 1101
AMS_00102 NODE_726_length_1395_cov_17.3326 100.00 1395
AMS_01465 NODE_718_length_1414_cov_20.2716 100.00 1413
AMS_01716 NODE_777_length_1178_cov_19.8879 100.00 1176
AMS_01785 NODE_771_length_1194_cov_19.8293 100.00 1194
AMS_02653 NODE_800_length_1062_cov_18.9562 100.00 1062
AMS_02811 NODE_720_length_1410_cov_17.9443 100.00 1410

Basically, all genes are assembled, but only on contigs which have exactly
the size of the HGT gene.

Comments for the author

The presented results are very surprising and the authors should elaborate on
the findings and try to interpret these in more detail. The manuscript would improve significantly from digging a little deeper into the results and trying to explain what has been observed.

A way to do this might be to load the metaSPAdes results into a tools like e.g. Bandage (https://rrwick.github.io/Bandage/) and visualize the HGT genes in the assembly graph.

I tried to re-run metaSPAdes, but I failed as some files were missing in the GitHub repository,
which made the presented results not reproducible (see above).

---

## Round 0.3 · accepted · Accept

We look forward getting your manuscript publication-ready.